# CCIM: Cross-modal Cross-lingual Interactive Image Translation

**Cong Ma**[1,2], **Yaping Zhang**[1,2]*, **Mei Tu**[4], **Yang Zhao**[1,2], **Yu Zhou**[2,3], **Chengqing Zong**[1,2]

[1]School of Artificial Intelligence, University of Chinese Academy of Sciences, Beijing, China
[2]State Key Laboratory of Multimodal Artificial Intelligence Systems (MAIS),
Institute of Automation, Chinese Academy of Sciences, Beijing, China
[3]Fanyu AI Laboratory, Zhongke Fanyu Technology Co., Ltd, Beijing, China
[4]Samsung Research China - Beijing (SRC-B)
{cong.ma, yaping.zhang, yang.zhao, yzhou, cqzong}@nlpr.ia.ac.cn
mei.tu@samsung.com

## Abstract

Text image machine translation (TIMT) which translates source language text images into target language texts has attracted intensive attention in recent years. Although the end-to-end TIMT model directly generates target translation from encoded text image features with an efficient architecture, it lacks the recognized source language information resulting in a decrease in translation performance. In this paper, we propose a novel Cross-modal Cross-lingual Interactive Model (CCIM) to incorporate source language information by synchronously generating source language and target language results through an interactive attention mechanism between two language decoders. Extensive experimental results have shown the interactive decoder significantly outperforms end-to-end TIMT models and has faster decoding speed with smaller model size than cascade models. [1]

## 1 Introduction

Text image machine translation (TIMT) aims at translating text in images from the source language into the target language, which has been widely used in various applications such as photo translation, scene text translation, digital document translation, and so on. Existing research on TIMT is mainly divided into two categories of methods: cascade method and end-to-end method. Cascade method (Hinami et al., 2021; Shekar et al., 2021; Afli and Way, 2016; Chen et al., 2015; Du et al., 2011) takes a text image recognition (TIR) model for source language text recognition (Baek et al.; Shi et al., 2017, 2016; Zhang et al., 2021, 2019) and then translates them into target language texts with a machine translation (MT) model (Vaswani et al., 2017; Gehring et al., 2017a,b; Johnson et al., 2017; Bahdanau et al., 2015; Sutskever et al., 2014; Zhao et al., 2019, 2020). To explicitly recognize the

source language embedded in text images, the cascade model combines TIR models and MT models for the TIMT task. However, two individual models in the cascade frame have double parameters and the decoding speed is slow. Meanwhile, errors in the TIR model are further propagated in the MT model leading to performance decrease. The end-to-end method directly translates the source language text image into target language through a unified encoder-decoder architecture, which is more parameter-efficient than cascade models with faster decoding speed (Ma et al., 2022; Su et al., 2021; Mansimov et al., 2020; Chen et al., 2020; Ma et al., 2023a,b,c).

However, the performance of end-to-end models is limited because the translation process lacks explicit source language guidance from recognition texts. An intuitive solution is to incorporate the recognition history into the translation decoder to offer more efficient guidance. Recently, multi-source interaction has been studied to incorporate effective information into target model (Lu et al., 2022; Xu et al., 2021; He et al., 2021; Liu et al., 2020; Zhou et al., 2019a,b; Wang et al., 2019; Zoph and Knight, 2016). Although multi-source interaction is vital to enhance the encoding capacity of TIMT model through attending recognition information explicitly, it has not been explored yet.

To address the above issues, we propose a novel Cross-modal Cross-lingual Interactive Model (CCIM) for TIMT, which effectively incorporates source language recognition information into the TIMT decoder through interactive attention. The interactive decoder has two decoding modules, one for source language and the other one for target language generation. A cross-lingual interactive attention mechanism is introduced to bridge the two language decoders. When generating translation results, the target language decoder not only receives the hidden states from the encoder and previous decoded translation history but also attends to the

---

*Corresponding author.
[1]https://github.com/EriCongMa/CCIM

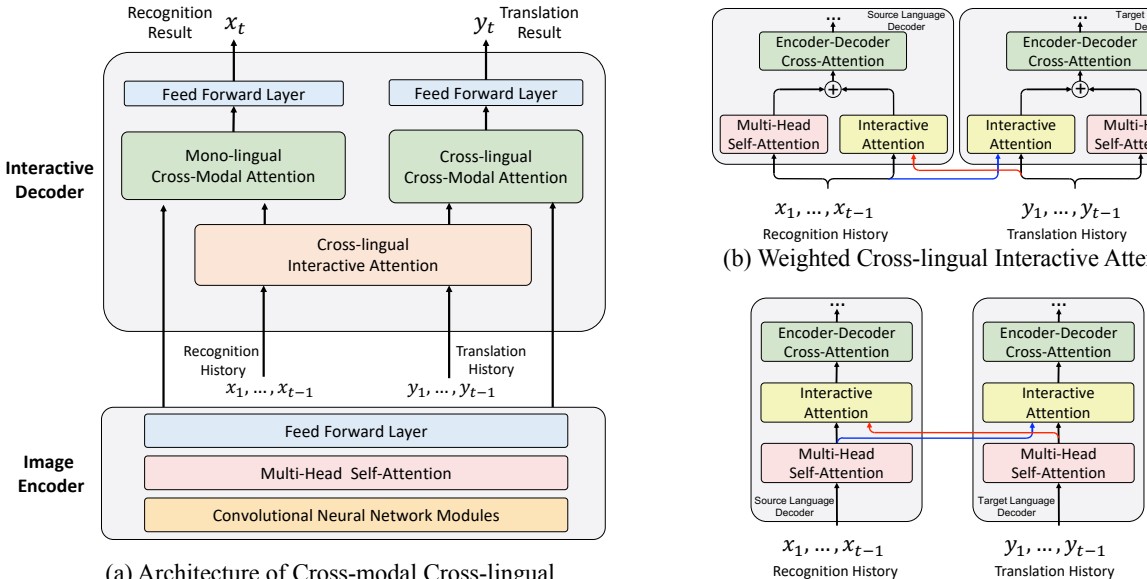

(a) Architecture of Cross-modal Cross-lingual Interactive Model for Text Image Translation

(b) Weighted Cross-lingual Interactive Attention

(c) Hierarchical Cross-lingual Interactive Attention

Figure 1: (a) illustrates the proposed cross-modal cross-lingual interactive model (CCIM). (b) illustrates the weighted cross-lingual interactive attention module. (c) illustrates the hierarchical cross-lingual interactive attention module.

decoded recognition history. Our contributions are summarized as follows:

- We propose a novel cross-modal cross-lingual interactive model (CCIM) for the TIMT task, which effectively enhances the translation decoder by incorporating recognition features.

- Weighted and hierarchical interactive decoding strategies have been studied to validate the effectiveness of interactive generation.

- Experimental results on three evaluation datasets have revealed the CCIM improves the translation quality of end-to-end TIMT models and outperforms cascade models with fewer parameters and faster decoding speed.

## 2 Methodology

### 2.1 Cross-modal Cross-lingual Interactive Model

The proposed CCIM model consists of an image encoder and an interactive decoder. As shown in Figure 1 (a), the image encoder first extracts image features given the source language text image, then two decoders are utilized for text image recognition and translation synchronously.

For image encoding, a convolutional neural network is utilized to extract image representation through multi-layer convolution and pooling operations (He et al., 2016). While for multi-head attention (MHA), the model collects information from different positions to update the hidden state of the current position (Vaswani et al., 2017):

$$\text{MHA}(Q, K, V) = \text{Concat}(\text{head}_1, ..., \text{head}_h)W_O$$
$$\text{where } \text{head}_i = \text{Attention}(QW_Q^i, KW_K^i, VW_V^i) \quad (1)$$

where $W_Q^i, W_K^i$ and $W_V^i$ represent query, key, and value projection matrices for head $i$, respectively. $W_O$ denotes the output projection matrix.

**Self-attention (SA)** The interactive decoder first calculates self-attention hidden states for both source language $X$ and target language $Y$ given the same query, key, and value:

$$H_X^{\text{SA}} = \text{MHA}(X, X, X)$$
$$H_Y^{\text{SA}} = \text{MHA}(Y, Y, Y) \quad (2)$$

Then, cross-lingual interactive-attention (IA) hidden states are calculated through two language decoders. Two types of IA mechanisms are utilized to recognize and translate synchronously:

**Weighted Interactive Attention (WIA)** As shown in Figure 1 (b), the self-attention and interactive attention are calculated separately and then weighted summation:

$$H_X^{\text{WIA}} = H_X^{\text{SA}} + \lambda \times \text{MHA}(X, Y, Y)$$
$$H_Y^{\text{WIA}} = H_Y^{\text{SA}} + \lambda \times \text{MHA}(Y, X, X) \quad (3)$$

where the query of WIA is from the corresponding language decoding history. Key and value are from the other language history.

**Hierarchical Interactive Attention (HIA)** To fuse self- and interactive-attention together, a hierarchical calculation mechanism is introduced to

| | Synthetic TIMT | | | Subtitle TIMT | Street-view TIMT | MT Dataset | TIR Dataset |
|---|---|---|---|---|---|---|---|
| | #Train | #Valid | #Test | #Test | #Test | #Train | #Train |
| Zh⇒En | 1,000,000 | 2,000 | 2,502 | 1,040 | 1,198 | 5,984,287 | 1,000,000 |
| En⇒Zh | 1,000,000 | 2,000 | 2,502 | 1,040 | - | 5,984,287 | 1,000,000 |
| En⇒De | 1,000,000 | 2,000 | 2,000 | - | - | 20,895,771 | 1,000,000 |

Table 1: Statistics of text image machine translation (TIMT), machine translation (MT), and text line image recognition (TIR) datasets.

obtain interactive information through serial computing as shown in Figure 1 (c):

$$H_X^{\text{HIA}} = \text{MHA}(H_X^{\text{SA}}, H_Y^{\text{SA}}, H_Y^{\text{SA}})$$
$$H_Y^{\text{HIA}} = \text{MHA}(H_Y^{\text{SA}}, H_X^{\text{SA}}, H_X^{\text{SA}}) \tag{4}$$

where the query is from the inner-lingual self-attention results, while the key and value are from the other language self-attention features.

**Encoder-Decoder Cross-Attention (CA)** Cross-lingual interactive-attention hidden states are fed into the encoder-decoder cross-attention mechanism to further incorporate encoder features into the decoder as in (Vaswani et al., 2017).

$$H_X^{\text{CA}} = \text{MHA}(H_X^{\text{IA}}, H_I, H_I)$$
$$H_Y^{\text{CA}} = \text{MHA}(H_Y^{\text{IA}}, H_I, H_I) \tag{5}$$

where $H_I$ represents the hidden states from the image encoder. $H_X^{\text{IA}}, H_Y^{\text{IA}}$ can be WIA or HIA for source and target languages. $H_X^{\text{CA}}, H_Y^{\text{CA}}$ denote the output of the encoder-decoder cross-attention module for source and target language, respectively.

The hidden states from the encoder-decoder cross-attention mechanism are then further encoded by the feedforward layer to obtain the interactive decoder layer outputs. Notes that the residual connections (He et al., 2016) and layer normalization (Ba et al., 2016) in standard transformer decoder are also utilized after self-attention, interactive attention, encoder-decoder cross attention and feedforward modules in interactive decoder (Zhao et al., 2023), which are not drawn in Figure 1 for simplification.

## 2.2 Loss Functions for Optimization

Since the interactive decoder has two decoders for the source language and target language respectively, TIR and TIMT tasks are optimized synchronously by multi-task learning. The training dataset contains triple paired samples as $D = \{I^i, X^i, Y^i\}_i^{|D|}$, where $I^i$ is the $i$-th source language text image, $X^i$ is the $i$-th source language texts and $Y^i$ is the corresponding translated target

language texts. The model is updated by optimizing both TIR and TIMT loss functions:

$$\mathcal{L} = \mathcal{L}_{\text{TIR}} + \mathcal{L}_{\text{TIMT}}$$
$$\mathcal{L}_{\text{TIR}} = -\sum_i^{|D|} \sum_j^{M} \log P(x_j^i | I^i, x_{<j}^i, y_{<j}^i)$$
$$\mathcal{L}_{\text{TIMT}} = -\sum_i^{|D|} \sum_j^{N} \log P(y_j^i | I^i, x_{<j}^i, y_{<j}^i) \tag{6}$$

where $x_{<j}$ and $y_{<j}$ denote the recognition and translation history. $M$ and $N$ represent the token length of the source language and target language. Note that the interactive decoder has three attention modules (self-attention, interactive attention from the other task, and encoder-decoder attention), the decoder generates tokens given the condition of both text image, recognition history, and translation history. Thus the interactive decoder has the potential to generate more accurate translation results.

## 2.3 Training and Inference

For each decoding step during training, the teacher-forcing decoding strategy is utilized to train the parameters in the decoder in a parallel computing way and each position in the decoder can attend to all positions in the decoder up to and including that position through the attention mask. During inference, the decoder generates both source and target language tokens by tokens in an auto-regressive way. For each step, the two sub-branches of the interactive decoder can attend encoder features, recognition history features, and translation history features, and predict both source and target language at the current step.

## 3 Experiments and Results

### 3.1 Datasets

The experiments have been conducted on a public TIMT corpus released by (Ma et al., 2022). The training set contains one million triple-paired samples of source language images, source language

| Architecture | Synthetic | | | Subtitle | | Street |
|---|---|---|---|---|---|---|
| | En⇒Zh | En⇒De | Zh⇒En | En⇒Zh | Zh⇒En | Zh⇒En |
| CLTIR (Chen et al., 2020) | 18.02 | 15.55 | 10.74 | 16.47 | 9.04 | 0.43 |
| +TIR | 19.44 | 16.31 | 13.52 | 17.96 | 11.25 | 1.74 |
| RTNet (Su et al., 2021) | 18.91 | 15.82 | 12.54 | 17.63 | 10.63 | 1.07 |
| +TIR | 19.63 | 16.78 | 14.01 | 18.82 | 11.50 | 1.93 |
| MTETIMT(Ma et al., 2022) | 19.25 | 16.27 | 13.16 | 17.73 | 10.79 | 1.69 |
| +MT | 21.96 | 18.84 | 15.62 | 19.17 | 12.11 | 5.84 |
| CCIM | **22.21** | **19.13** | **15.72** | **19.48** | **12.12** | **5.88** |

Table 2: Performance of end-to-end models. All end-to-end models are trained with the same TIMT training dataset. External TIR and MT corpus are also kept the same among different architecture settings.

texts, and target language translation pairs for each translation direction. The source language text images in the training dataset are synthesized by using bilingual text sentences. To validate the generalization of models, one synthetic test set and two real-word (subtitle and street-view) test sets are utilized to evaluate the translation performance. The statistics of the dataset are shown in Table 1.

## 3.2 Experimental Settings

Image encoder in CCIM utilizes the same configuration in (Ma et al., 2022). The source language and target language decoder are 6-layer transformer decoder with 512-dimensional hidden sizes as in (Vaswani et al., 2017; Zhao et al., 2023). The maximum sentence length for English, German, and Chinese are set to 80, 80, and 40 respectively. The preprocessed image height is set to 32 and the channel is 3. To align the length of the image feature and text feature, preprocessed image width is resized to 320, 320, and 160. The batch size is set to 64, and the training step is 300,000. All models are initialized with Xavier initiation method (Glorot and Bengio, 2010) and optimized with Adam optimizer (Kingma and Ba, 2015) on a single NVIDIA V100 GPU. Sacre-BLEU[2] (Papineni et al., 2002) is utilized for evaluation metric.

## 3.3 Baseline Models

- CLTIR model is a vanilla multi-task learning based TIMT model with auxiliary TIR task training (Chen et al., 2020).
- RTNet bridges the TIR encoder and MT decoder through a feature transformer, which is also trained with TIR task (Su et al., 2021).
- MTETIMT is a machine translation enhanced TIMT model, which is trained with both auxiliary TIR and MT tasks (Ma et al., 2022).

[2]https://github.com/mjpost/sacrebleu

| Architecture | BLEU↑ | Param.↓ | Speed↑ |
|---|---|---|---|
| Cascade | 20.46 | 195M | 3.07 |
| Our work: CCIM | 22.21 | 147M | 5.04 |

Table 3: Comparison of cascade and end-to-end CCIM models. The unit for parameters is million ($\times 10^6$), while the unit for speed is sentence per second.

## 3.4 Comparison with Different End-to-End TIMT Models

Table 2 shows the main results on three evaluation domains. As shown in Table 1, CCIM outperforms the existing best multi-task based MTETIT by 0.21 BLEU scores on average. Meanwhile, CCIM improves the translation performance on real-world domains by 0.12 BLEU scores on average, indicating the good generalization of our proposed method. Furthermore, CCIM can generate source language and target language synchronously, which can meet the requirement of both recognition and translation tasks in practical applications.

## 3.5 Model Size and Decoding Speed

The Cascade model deploys TIR and MT models, leading to parameter redundancy and decoding delay. With an end-to-end architecture, CCIM outperforms the cascade model with fewer parameters and faster decoding speed as shown in Table 3. Specifically, CCIM decreases around 24.62% parameters and achieves 1.64x acceleration compared with the cascade model. Meanwhile, CCIM significantly outperforms the cascade model by 1.75 BLEU scores, which effectively alleviates the error propagation problem in the cascade model.

## 3.6 Comparison of Different Interactive Attention Types

To validate the effectiveness of interactive attention, an ablation study of replacing key and value

| Interactive Attention Type | BLEU |
|---|---|
| Weighted Interactive Attention (Rand) | 8.07 |
| Hierarchical Interactive Attention (Rand) | 11.23 |
| Weighted Interactive Attention (WIA) | 22.94 |
| Hierarchical Interactive Attention (HIA) | 24.18 |

Table 4: Comparison of Various Interactive Attention Types on English-to-Chinese validation set.

| | |
|---|---|
| Recognition Ground Truth (Pinyin) | 我们 需要 再次 排查 流程 (women xuyao zaici paicha liucheng) |
| End-to-end TIT | We need to check the process |
| Multi-task | We need to check the process |
| CCIM | We need to double-check the process |
| Translation Ground Truth | We need to double check the process |

Figure 2: Case study of end-to-end TIMT models.

in interactive attention with random samples noise vector has been implemented. Experimental results in Table 4 show that random noise replaced interactive attention generates a poor translation, especially for weighted interactive attention. We attribute that weighted interactive attention incorporates noise signals through weighted summation, which severely disturbs the information flow. Furthermore, hierarchical interactive attention outperforms weighted interactive attention, which reveals that flexible calculation of hierarchical architecture is better than vanilla summation operation.

### 3.7 Case Study of CCIM Model

Fig. 2 shows an example of TIMT generated by end-to-end and CCIM models. Although the end-to-end model translates the general meaning of the sentence, it ignores the meaning of 'double-check' in the source language text image. Since there is no interaction during decoding, the multi-task based model also ignores this meaning. CCIM successfully translated this word through interactive attention with the source language decoder, indicating CCIM can effectively alleviate the problem of lacking source language information in vanilla end-to-end TIMT models.

### 3.8 Wait-$k$ Strategy for CCIM

The wait-$k$ strategy is commonly employed in speech translation, aiming at generating better translation given more recognition history. To validate the wait-$k$ strategy in the TIMT task, we also conducted corresponding experiments as shown in Table 5. From the experimental result, the wait-$k$

| Wait-$k$ | WIA | HIA |
|---|---|---|
| Wait-0 | 22.94 | 24.18 |
| Wait-1 | 22.99 | 24.32 |
| Wait-2 | 23.07 | 24.75 |
| Wait-3 | 23.64 | 24.91 |
| Wait-4 | **23.75** | **25.49** |
| Wait-5 | 23.36 | 25.18 |

Table 5: The performance of wait-$k$ strategy for CCIM on English-to-Chinese Validation Set.

strategy makes the recognition task decode first, enabling the translation task to access more source language information for improved translation quality. While the wait-$k$ strategy enhances translation quality, it does introduce some latency increase. The CCIM model achieves the best translation performance when $k = 4$ in our experiments.

### 4 Conclusion

This paper proposes a novel interactive decoder based end-to-end TIMT model, which explicitly incorporates recognized hidden states into the translation process. Through the interactive attention mechanism, recognition and translation results are generated synchronously and mutually enhanced. By making full use of the source language recognition information, CCIM outperforms existing end-to-end and multi-task based TIMT models on both synthetic and real-world evaluation sets. Furthermore, with the end-to-end architecture, CCIM has fewer parameters and faster decoding speed than cascade models. Ablation study of different interactive attention types shows hierarchical interactive attention has stronger interactive ability across recognition and translation tasks. In the future, we will explore more interactive methods for end-to-end text image machine translation.

### 5 Acknowledgements

This work has been supported by the National Natural Science Foundation of China (NSFC) grants 62106265.

### 6 Limitations

Our method is now designed for text line images, which need preprocessing of text detection in images. In the future, we will consider optimizing the text detection and translation in images jointly to increase the scalability of our work.

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
