# OpenReview forum: "CCIM: Cross-modal Cross-lingual Interactive Image Translation"
_EMNLP/2023/Conference — EMNLP 2023 Findings_

### Official Review · Reviewer_gHG5 · 2023-07-24

**Soundness:** 3

**Excitement:**

3: Ambivalent: It has merits (e.g., it reports state-of-the-art results, the idea is nice), but there are key weaknesses (e.g., it describes incremental work), and it can significantly benefit from another round of revision. However, I won't object to accepting it if my co-reviewers champion it.

**Paper Topic And Main Contributions:**

This paper introduces interactive decoding into Text Image Machine Translation (TIMT), enabling synchronous generation of source and target sentences. Results on the recently proposed benchmark demonstrate that the proposed method, CCIM, outperforms both the cascaded baseline and previous TIMT models.

**Questions For The Authors:**

Have you tried the wait-k decoding policy in interactive decoding as proposed in previous works?

**Reasons To Accept:**

- The proposed method achieves promising results on the TIMT benchmark.
- The writing is clear and easy to follow.

**Reasons To Reject:**

This paper directly applies interactive attention to the TIMT task, similar to its previous applications in speech translation (Liu et al., 2020) and multilingual NMT (He et al., 2021). As a result, the novelty of the proposed method is somewhat limited.

**Reproducibility:**

4: Could mostly reproduce the results, but there may be some variation because of sample variance or minor variations in their interpretation of the protocol or method.

**Reviewer Confidence:**

4: Quite sure. I tried to check the important points carefully. It's unlikely, though conceivable, that I missed something that should affect my ratings.

---

> ### Author Rebuttal · Authors · 2023-08-29
>
> We appreciate your valuable time in reviewing our paper with the acknowledgment of our experimental results and paper writing. For your suggestions and questions, we address the most crucial details as follows:
>
> Q1. On the novelty of the proposed cross-lingual cross-modal interactive model.
>
> A1. In fact, Compared to the interactive decoding method in speech translation and multilingual machine translation, the proposed CCIM not only blended it into the TIMT task but also innovatively introduced the hierarchical interactive attention mechanism. First, in our paper, experimental results on various translation directions and domains prove the interactive decoding strategy is effective in the TIMT task. Second, different from the previous weighted sum interactive attention mechanism in speech translation or multilingual machine translation, the proposed novel hierarchical interactive attention mechanism unifies the source and target language decoding progress and jointly calculates the hidden states in decoding, which has been proven more effective than previous works in Table 1. Therefore, we sincerely hope that the reviewers can consider raising the score for "Excitement". Moreover,  Table 3 shows that the hierarchical interactive attention mechanism outperforms other interactive attention methods with 1.24 BLEU highlighting the hierarchical interactive attention mechanism is effective and more suitable for the TIMT task. Thus, we sincerely hope reviewers to kindly consider increasing the score for "Soundness" as well.
>
> Q2. On the consideration of the wait-k decoding policy in interactive decoding.
>
> A2. Thanks for your suggestion. Indeed, the wait-k strategy is commonly employed in speech translation, and we also conducted corresponding experiments in the TIMT task as well. From the experimental result, the wait-k strategy makes the recognition task decode first, enabling the translation task to access more source language information for improved translation quality. While the wait-k strategy enhances translation quality, it does introduce some latency increase. We will provide a detailed analysis experiments of the trade-off between translation quality and decoding latency in the revised version. Thanks again for your valuable suggestion to improve the analysis part of CCIM.

---

### Official Review · Reviewer_jT2T · 2023-08-05

**Soundness:** 4

**Excitement:**

3: Ambivalent: It has merits (e.g., it reports state-of-the-art results, the idea is nice), but there are key weaknesses (e.g., it describes incremental work), and it can significantly benefit from another round of revision. However, I won't object to accepting it if my co-reviewers champion it.

**Paper Topic And Main Contributions:**

This paper aims at Text image machine translation (TIMT) and proposes an architecture to model visual features and cross-lingual features. Specifically, a cross-lingual cross-modal attention is used to capture image context, and a cross-lingual interactive attention is used to capture cross-lingual text context. The loss consists of both original language recognization and cross-lingual translation. Experiments show the effectiveness.

**Questions For The Authors:**

1. What's the version of Cross-lingual Cross-Modal Attention, weighted or hierarchical. Have you tried the other way?
2. Are the results of other work from original paper or the author re-implement them?

**Reasons To Accept:**

1. The motivation is clear. The methodology is clearly illustrated with well-drawing figures.
2. Well comparison like with cascade and other work.
3. Attempts with Weighted Cross-lingual Interactive Attention and Hierarchical Cross-lingual Interactive Attention.


**Reasons To Reject:**

1. More related work should be introduced more.
2. More analysis should be conducted, such as what does those attention represent? Can they be visualized?

**Reproducibility:**

4: Could mostly reproduce the results, but there may be some variation because of sample variance or minor variations in their interpretation of the protocol or method.

**Reviewer Confidence:**

4: Quite sure. I tried to check the important points carefully. It's unlikely, though conceivable, that I missed something that should affect my ratings.

---

> ### Author Rebuttal · Authors · 2023-08-29
>
> We appreciate your valuable time in reviewing our paper with the acknowledgment of our motivation and comparison experiments. For your suggestions and questions, we address the most crucial details as follows:
>
> Q1. On the more introduction of related work.
>
> A1. Thanks for your nice suggestion for the related work part. Following your suggestions, we assure you that we will add more details and analysis of related work in our revised version.
>
> Q2. On the details of more analysis such as attention representation and visualization.
>
> A2. Thanks for your suggestions on the clarification of interactive attention. We will add an analysis section on interactive attention to deeply analyze the effectiveness of interactive attention representation. Meanwhile, we will also follow your suggestion and visualize the interactive attention map to explain the reason behind the effectiveness of our proposed interactive method for the TIMT task.
>
> Q3. What's the version of Cross-lingual Cross-Modal Attention, weighted or hierarchical? Have you tried the other way?
>
> A3. The version of cross-lingual cross-modal Attention in Table 1 and Table 2 is the hierarchical interactive attention. We also conduct analysis experiments on the English-to-Chinese validation set as shown in Table 3 and the results show that hierarchical interactive attention outperforms weighted interactive attention. As a result, we evaluate the hierarchical interactive attention-based CCIM on the test set to compare with existing related work.
>
> Q4. Are the results of other work from the original paper or did the author re-implement them?
>
> A4. The experimental results we report for related work are sourced from the published results in [Ma et al., 2022]. We also re-produced the models in Table 1 and there are minor differences between the re-produced results and the original results in the paper. Thus we reported the original results in the paper to make it consistent with the results in published related work.

---

### Official Review · Reviewer_DKy3 · 2023-08-05

**Soundness:** 3

**Excitement:**

3: Ambivalent: It has merits (e.g., it reports state-of-the-art results, the idea is nice), but there are key weaknesses (e.g., it describes incremental work), and it can significantly benefit from another round of revision. However, I won't object to accepting it if my co-reviewers champion it.

**Paper Topic And Main Contributions:**

This paper studies the text image machine translation, which translates the source sentence in the image into the target language text. The authors propose a novel interactive decoder that generates the source and target language texts in interactive decoding strategies.

**Reasons To Accept:**

1.  The paper proposes a novel approach to improve text image machine translation.

2. Experimental results demonstrate the effectiveness of the proposed approach.

**Reasons To Reject:**

1. The most essential parts of the method are not described in detail.  Eq（6）in Section 2.2 (Loss Functions for Optimization) and Figure 1  indicate that the interactive decoder generates a source token and a target token from the hidden states at each decoding step.

However, for the machine translation task, the interactive decoding strategy faces the following challenges: 1) The length of the source sentence is different from that of the target sentence. 2) Word reordering between the source and target sentences.

**Reproducibility:**

4: Could mostly reproduce the results, but there may be some variation because of sample variance or minor variations in their interpretation of the protocol or method.

**Reviewer Confidence:**

3: Pretty sure, but there's a chance I missed something. Although I have a good feel for this area in general, I did not carefully check the paper's details, e.g., the math, experimental design, or novelty.

---

> ### Author Rebuttal · Authors · 2023-08-29
>
> We appreciate your valuable time in reviewing our paper with the acknowledgment of our novelty and experiments. Thank you for your concern on more details of our methods. For easier reproducibility, we will release our code to the public. Here, we address the most crucial details:
>
> Q1. On the details of Eq (6) in Section 2.2 (Loss Functions for Optimization).
>
> A1. Thanks for your consideration of the details of our work. Equation (6) shows the overall loss function by calculating both recognition and translation losses. Since the interactive decoder has 3 attention modules (self-attention, interactive attention from the other task, and encoder-decoder attention), the decoder generates tokens given the condition of both text image, recognition history, and translation history. Our code will be released to the public as introduced in the abstract to ensure good reproducibility. We will also add more technical details in our revised version to help readers understand the details of our work.
>
> Q2. On the details of each decoding step.
>
> A2. For each decoding step during training, the teacher-forcing decoding strategy is utilized to train the parameters in the decoder in a parallel computing way and each position in the decoder can attend to all positions in the decoder up to and including that position through the attention mask. During inference, the decoder generates both source and target language tokens by tokens in an auto-regressive way. For each step, the two sub-branches of the interactive decoder can attend encoder features, recognition history features, and translation history features, and predict both source and target language at the current step. The difference between weighted cross-lingual interactive attention and hierarchical cross-lingual interactive attention is the calculation method of the interactive attention. The hierarchical cross-lingual interactive attention is utilized as the final version of CCIM in Tables 1 and 2. Our code will be released to the public as introduced in the abstract to ensure good reproducibility. We will also add more technical details in our revised version to help readers understand the details of our work.
>
> Q3. On the consideration of the length difference between source and target sentences.
>
> A3. The source and target sentences indeed have different lengths. During the progress of decoding, an end-of-sentence <EOS> token is utilized as the symbol of the end of the sentence. For batch decoding, all the sentences no matter the source language and target language are supposed to generate tokens up to maximum length except for all samples that have met the <EOS> token. When the source sentence has generated <EOS> and the target language has not, the decoding procedure is continued and the interactive attention mask will mask the positions after <EOS> token in the source language so that the features at the new decoding position will attend all the decoded history and ignores the unuseful features after <EOS> position, vice versa. We will also add more technical details in our revised version to help readers understand the details of our work.
>
> Q4. On the consideration of word reordering between the source and target sentences.
>
> A4. Source and target sentences have different word orders, and this is one of the reasons that the interactive decoding strategy is useful. With the different orders of source and target sentences, both the source and target language decoder can attend to more information except for text image encoder features, thus the generation procedure can generate more accurate tokens at each step. We will add a case study about reordering analysis to help readers understand the details of our work.
>
> We greatly appreciate your attention to the details and experimental details. We assure you that our revised version will incorporate more details to enhance readability and our code will be released to the public. Our proposed method introduces novel interactive attention for the TIMT task and significantly improves the experimental results. Reviewer #2 and Reviewer #3 both have a high acknowledgment of our work, therefore we sincerely hope you could kindly consider raising the scores for "soundness" and "excitement".

---

### Meta-Review · Area_Chair_Gbpr · 2023-09-19

**Recommendation:** 3

**Metareview:**

This paper proposes an interactive image translation method that generates source and target language tokens jointly as a multi-task problem using cross-lingual and cross-modal attention mechanisms.
The reviewers agree with the clarity of the paper and the effectiveness of the proposed method.
There was a discussion on the novelty of the approach, but it seems to be resolved by the authors’ rebuttal.

Here is a list of the pros and cons of this paper.
* Pros
- Clear and well-written (jT2T, gHG5)
- Effectiveness against existing methods (DKy3, jT2T, gHG5)
* Cons
- Some essential technical details should be improved (DKy3)
- More analyses should be conducted (e.g., visualization of attentions) (jT2T)

---

### Decision · Program_Chairs · 2023-10-07

**Decision:**

Accept-Findings

**Comment:**

This paper proposes an interactive image translation method that generates source and target language tokens jointly as a multi-task problem using cross-lingual and cross-modal attention mechanisms.
The reviewers agree with the clarity of the paper and the effectiveness of the proposed method.
There was a discussion on the novelty of the approach, but it seems to be resolved by the authors’ rebuttal.

Here is a list of the pros and cons of this paper.
* Pros
- Clear and well-written (jT2T, gHG5)
- Effectiveness against existing methods (DKy3, jT2T, gHG5)
* Cons
- Some essential technical details should be improved (DKy3)
- More analyses should be conducted (e.g., visualization of attentions) (jT2T)